# Global Epidemiology and Burden of Elderly-Onset Inflammatory Bowel Disease: A Decade in Review

**DOI:** 10.3390/jcm12155142

**Published:** 2023-08-06

**Authors:** Pojsakorn Danpanichkul, Kanokphong Suparan, Suchapa Arayakarnkul, Aunchalee Jaroenlapnopparat, Natchaya Polpichai, Panisara Fangsaard, Siwanart Kongarin, Karan Srisurapanont, Banthoon Sukphutanan, Wasuwit Wanchaitanawong, Yatawee Kanjanakot, Jakrapun Pupaibool, Kwanjit Duangsonk, Gursimran Singh Kochhar, Karn Wijarnpreecha

**Affiliations:** 1Immunology Unit, Department of Microbiology, Faculty of Medicine, Chiang Mai University, Chiang Mai 50200, Thailand; 2Department of Medicine, University of Minnesota, Minneapolis, MN 55455, USA; 3Department of Medicine, Mount Auburn Hospital, Beth Israel Lahey Health, Cambridge, MA 02138, USA; 4Department of Internal Medicine, Weiss Memorial Hospital, Chicago, IL 60640, USA; 5Department of Internal Medicine, Bassett Medical Center, Cooperstown, NY 13326, USA; 6Faculty of Medicine, Chiang Mai University, Chiang Mai 50200, Thailandbanthoonsuk@gmail.com (B.S.); 7Division of Gastroenterology and Hepatology, Department of Internal Medicine, Faculty of Medicine, Chiang Mai University, Chiang Mai 50200, Thailand; 8Department of Surgery, School of Medicine, Mae Fah Luang University, Chiang Rai 57100, Thailand; 9Division of Infectious Diseases, Department of Internal Medicine, University of Utah, Salt Lake City, UH 84112, USA; 10Department of Microbiology, Faculty of Medicine, Chiang Mai University, Chiang Mai 50200, Thailand; 11Division of Gastroenterology and Hepatology, Allegheny Health Network, Pittsburgh, PA 15212, USA; 12Division of Gastroenterology and Hepatology, Department of Medicine, University of Arizona College of Medicine, Phoenix, AZ 85004, USA; 13Division of Gastroenterology and Hepatology, Department of Internal Medicine, Banner University Medical Center, Phoenix, AZ 85006, USA

**Keywords:** inflammatory bowel disease, Crohn’s disease, ulcerative colitis, geriatric

## Abstract

Inflammatory bowel disease (IBD), once thought to impact younger individuals, now manifests in approximately 10% of patients over 65, characterized by a heightened vulnerability to complications and greater diagnostic intricacies than conventional cases. However, comprehensive global epidemiological data regarding elderly-onset IBD are currently insufficient. Our study addresses this critical gap by analyzing trends in elderly-onset IBD over a decade, encompassing the estimation of annual frequencies and age-standardized rates of elderly-onset IBD burden for both genders, stratifying the data by geographical and sociodemographic factors. Our research highlights a notable increase in the proportion of elderly-onset IBD, constituting around 13% of all IBD cases. We observed a rising incidence in males, contrasted by a decreasing trend in females. The highest surge in incidence rates was seen in the Western Pacific region in both genders, but the highest burden was observed in America. Countries with high sociodemographic index (SDI) carried the greatest burden of elderly-onset IBD, while countries with low SDI had the least. The mortality and disability-adjusted life years (DALYs) rates trend downward in most regions. This study underscores an increasing incidence and proportion of IBD, particularly in elderly-onset IBD, particularly in males. While mortality and DALYs are decreasing in most regions, the overall burden remains highest in America and high-SDI countries. Effective public health interventions and comprehensive studies are required to tackle this mounting burden.

## 1. Introduction

Inflammatory bowel disease (IBD) encompasses chronic inflammatory conditions affecting the gastrointestinal tract, including Crohn’s disease and ulcerative colitis [1,2]. Historically, IBD was primarily found in Western countries such as North America and Europe. However, in the 21st century, there has been a global surge in IBD incidence, particularly in newly industrialized countries in Asia, South America, and Africa, where Westernization has been established [3].

The incidence and prevalence of IBD are increasing worldwide, posing a significant challenge to healthcare systems due to the complexity of the disease and the high cost of healthcare utilization [4]. Although IBD is considered to affect primarily younger patients, a growing body of evidence shows a shift toward older age, possibly due to aging and increased exposure to environmental factors including, but not limited to, Western diet and industrialization [5,6,7]. The elderly population with IBD has multiple unique features, including a high risk of infection and malignancy complications, suboptimal surgical outcomes, and a high hospitalization rate compared to the younger population [8,9,10]. The presence of atypical presentations, comorbidities, and polypharmacy in elderly adults also necessitates tailored approaches [11,12]. Currently, no consensus exists to guide the management of this vulnerable group, possibly due to elderly patients often being excluded from therapeutic trials and the paucity of data regarding this entity [13,14]. Using the Global Burden of Disease Study, our research primarily aims to fill the gap in the literature by assessing temporal change in elderly-onset IBD spanning over a decade.

## 2. Materials and Methods

This research utilized information from the Global Burden of Disease Study 2019 (GBD 2019). This initiative was a comprehensive global undertaking that strived to assess the impact of diseases and risk factors in 204 countries and territories. The study investigated the occurrence, mortality rates, and disability-adjusted life years (DALYs) related to elderly-onset IBD globally using data obtained from the online data catalog query tool, known as the Global Health Data Exchange (GHDx), which the Institute for Health Metrics and Evaluation maintains.

Elderly-onset IBD is defined as IBD diagnosed in patients aged 65–89 years, and the International Classification of Diseases-10 (ICD-10) codes K50 (Crohn’s disease) and K51 (ulcerative colitis) were used to map the presence of elderly-onset IBD. The primary objective of this study was to evaluate the occurrence of elderly-onset IBD worldwide from 2010 to 2019 in six regions, according to the World Health Organization (WHO), including Africa, Eastern Mediterranean, European, the region of Americas, Southeast Asia, and the Western Pacific. In addition to our primary analysis, our study also stratified countries based on the sociodemographic index (SDI), which is a composite measure that combines the ranking of per capita incomes, educational attainment, and fertility rates across various countries and territories (Appendix A). The SDI is rated on a scale of 0 to 1; a 0 signifies the minimum level of development regarding health, and a 1 represents the maximum theoretical level. The full methodology used to estimate the disease burden of IBD in GBD 2019 has been described [15]. The GBD 2019 study evaluated the quality of the data from each country/territory utilizing a rating scale of 0 (the lowest quality) to 5 (the highest quality). Data quality ratings for each country are demonstrated in Appendix A. Different statistical techniques, such as misclassification correction, garbage code redistribution, and noise reduction algorithms, were used to reduce the data heterogeneity. We employed the Cause of Death Ensemble model, a Bayesian geospatial regression analysis, to estimate the mortality linked to IBD by age, sex, country/territory, and year. The incidence of IBD was calculated using a mortality-to-incidence ratio. A subgroup analysis was conducted to examine gender differences. Incidence rate pertains to the number of newly identified cases of a particular disease within a specific population during a specific period, divided by the size of the population at risk within the same period.

Estimates for the frequency of incident cases and deaths were reported with 95% uncertainty intervals (UIs) as 2.5th and 97.5th ranked values across all 1000 draws from a posterior distribution. Age-standardized incidence rate (ASIR) is calculated by dividing the number of new disease cases in a particular age group by the total population in that age group and multiplying it by 100,000, expressing the incidence as a population unit. Age-standardized death rate (ASDR) is obtained by dividing the number of deaths in a particular age group by the corresponding population of that age group and multiplying it by 100,000. In a similar manner, the number of age-standardized disability-adjusted life years (ASDALYs) is obtained by dividing the number of disabilities in a particular age group by the population of the specific age group and multiplying it by 100,000. The number of DALYs measures the burden of disease by combining years of life lost and years lived with disability. The number of years lost in life is determined by subtracting the age at death from the standard life expectancy. In contrast, the years lived with disability are computed by multiplying the number of individuals with a particular health condition by the average duration of that condition and a disability weight that reflects the severity of the disease. To assess the percentage change in any category between 2010 and 2019, the difference between the two years was divided by the initial value in 2010. The Joinpoint Regression Program, version 4.9.1.0 (Statistical Research and Applications Branch, National Cancer Institute), was utilized to estimate the annual percentage change (APC) and corresponding 95% confidence intervals (CIs) for the temporal modification in age-standardized rates (ASRs) between 2010 and 2019. An ascending trend was determined when both the APC and the lower boundary of the 95% CI were positive, whereas a descending trend was established when both the APC and the upper boundary of the 95% CI were negative.

## 3. Results

### 3.1. Incidence of Elderly-Onset IBD

In 2019, a total of 213,814 new cases of IBD were reported globally in males, out of which 26,029 (12.17%) were cases of elderly-onset IBD. In females, there were 190,738 new cases of IBD, with 27,298 (14.31%) being elderly-onset IBD cases (Table 1). The proportion of new IBD cases in the elderly to overall new IBD cases has an upward trend in all regions, with the Americas exhibiting the most substantial upward trend. Male cases increased from 12.25 to 15.73% (+3.48%), while female cases increased from 13.83 to 16.88% (+3.05%) (Figure 1, Table 1). In 2019, the ASIR was higher in males across all regions and all SDI strata (Figure 2A). The temporal progression of elderly-onset IBD from 2010 to 2019 showed that the ASIR significantly increased for males (APC: 0.29%, 95% CI: 0.21 to 0.36%, Figure 2B), and decreased for females (APC: −0.21%, 95% CI: −0.32 to −0.09%, Figure 2C) globally. The progression was most pronounced in males from the Western Pacific region (APC: 1.22%, 95% CI: 0.21 to 0.36%, Table 2), whereas males from Europe showed the smallest increase (APC: 0.14%, 95% CI: 0.03 to 0.25%). For females, the overall decline masked the regional variations (Figure 2B). The ASIR remained constant in the Americas and Europe, while all other regions experienced an upward trend, most notably Africa, where the female ASIR increased substantially (APC: 1.28%, 95% CI: 1.14 to 1.42%). The region of America had the highest ASIR in 2019 for both males and females, with 18.76 (95% UI: 15.54 to 22.28) and 17.31 (95% UI: 14.5 to 20.76) per 100,000, respectively. When stratified by SDI, males exhibited a greater increase in incidence across all SDI strata. Countries with a middle SDI displayed the highest increase in males’ (APC: 1.31%, 95% CI: 1.21 to 1.41%, Table 2) and females’ incidence (APC: 1.32%, 95% CI:1.24 to 1.39%). Nevertheless, countries with a high SDI had the highest ASIR in males and females, with an ASIR of 17.24 (95% UI: 14.24 to 20.57) and 15.13 (95% UI: 12.64 to 18.07) per 100,000, respectively. In a country-level analysis, ASIRs ranged from 0.49 (95% UI: 0.36 to 0.65) in Thailand to 33.34 (95% UI 30.99 to 35.94) per 100,000 in Canada. In addition to Canada, the countries with high ASIRs were Greenland, Sweden, and the United States. Greenland had an ASIR of 31.21 (95% UI: 24.28 to 29.2), Sweden had an ASIR of 29.53 (95% UI: 24.5 to 25.31), and the United States had an ASIR of 28.87 (95% UI: 23.74 to 34.82) per 100,000. The variation in the ASIRs in different countries is illustrated in Figure 3A and Appendix A.

### 3.2. Mortality and Disabilities of Elderly-Onset IBD

The estimated counts of death, DALYs, and rates (ASDRs and ASDALYs) of elderly-onset IBD can be found in Table 3 and Table 4 and Figure 2. Regarding mortality, the results indicate a downward trend in the worldwide ASDR for IBD among the elderly, both for males (APC: −1.19%, 95% CI: −1.29 to −1.08%, Figure 2E) and females (APC: −2.02%, 95% CI: −2.26 to −1.79%, Figure 2F). The decline in ASDR was more remarkable in females than males in all regions, with the most significant decreases observed in the Americas (APC; −1.56, 95% CI: −1.89 to −1.23%, Table 3). Europe exhibited the highest ASDR for both males and females, with an ASDR of 6.06 (95% UI: 4.47 to 6.75) and 6.41 (95% UI: 5 to 7.24) per 100,000, respectively. When stratified by SDI, the APC in females exceeded that of males in all groups, except for low-middle-SDI countries. In 2019, the highest ASDR was found within high SDI strata, with rates of 5.73 (95% UI: 4.21 to 6.39) in males and 6.9 (95% UI: 5.45 to 7.74) per 100,000 in females (Table 3). 

The change in DALYs mirrored the same trend observed in the mortality rates. The findings indicate a decrease in global ASDALYs for IBD among the elderly for both males (APC: −0.93%, 95% CI: −0.99 to −0.87%, Figure 2H) and females (APC: −1.4%, 95% CI: −1.54 to −1.26%, Figure 2I). ASDALYs for IBD among the elderly. In consideration of the WHO classification, all regions witnessed a decrease in DALYs, except for Europe (Table 4). The Western Pacific region displayed the most pronounced downward trend for both males (APC: −1.3%, 95% CI −1.5 to −1.1%) and females (APC: −2.35%, 95% CI −2.55 to −2.15%). Nevertheless, in a similar manner to ASDR, Europe exhibited the highest ASDALYs in 2019 for both genders, with ASDALYs of 117.92 (95% UI: 94.52 to 135.71) in males and 110.91 (95% UI: 92.03 to 125.59) per 100,000 in females. The differences in the ASDALYs across various countries are depicted at the end in Figure 3B. When stratified by SDI, countries with high SDI recorded the highest ASDALYs in 2019, with ASDALYs of 114.53 (95% UI: 90.7 to 131.47) in males and 122.49 (95% UI: 102.8 to 138.25) per 100,000 in females, respectively (Table 4). At the national level, ASDALYs varied significantly. Sri Lanka presented the lowest rate of 11.44 (95% UI: 8.35 to 15.7) per 100,000 people, while Germany had the highest at 33.34 (95% UI: 190.5 to 278.51) per 100,000. Other nations with the highest ASDALYs included the Netherlands, with a rate of 201.31 (95% UI: 164.27 to 231.71), the United Kingdom at 166.77 (95% UI: 143.44 to 186.46), and the United States at 150.31 (95% UI: 123.45 to 169.36) per 100,000. The differing ASDALYs across countries can be seen in Figure 3B and Appendix A.

## 4. Discussion

This study marks the first comprehensive examination of decade-long trends in incidence, deaths, and DALYs related to elderly-onset IBD on a global, regional, and national scale, drawing upon data from the GBD 2019 database. This database represents the most current and complete global resource on elderly IBD, and it provides more accurate estimations of disease burden than GBD 2017 database, due to refined case definitions and improved model fit incidence. 

Our study aligns with previous literature indicating a heterogeneous rise in the incidence rates of elderly-onset IBD in most countries, predominantly among males [12,16]. The most pronounced uptrend was observed in the Western Pacific region. With this rate of increment, the prevalence of IBD would possibly steadily climb over the next generation, similar to the rising prevalence observed in Western countries during the twentieth century [17]. However, the region with the highest incidence was still the region of the Americas. These overall uptrends masked significant regional variations, which showed an increasing burden in all regions but plateauing in the Americas and Europe. Regional disparities in the burden of IBD could be explained by varying risk factors, different database capture systems, and differing access to health care. In the SDI-stratified analysis, our study revealed large discrepancies in elderly-onset IBD burden in different SDI locations, with high-SDI countries exhibiting the highest burden. These findings were consistent with findings in the general IBD population, which attributed the burden to a higher extent of industrialization in high-SDI countries [3,18]. Of note, our study also demonstrated an increasing proportion of elderly-onset IBD compared to overall IBD. The cause of this second peak could be secondary to cumulative exposure to environmental factors such as smoking, Western diet, vitamin D deficiency, and obesity or the false positive rising attributed to misdiagnosis with other forms of colitis mimicking IBD in the elderly [19]. However, this emerging trend necessitates more thorough future studies. While our study underscored an upward trend in incidences, it also highlighted a significant decline in deaths and DALYs across most regions. The finding of our investigation does not establish a clear association between SDI and the extent of IBD-related deaths and DALYs. This may be partly attributed to the advancement of healthcare combined with economic and societal factors, which collectively transcend the classification of countries based on SDI.

The combined burden of elderly-onset IBD and adult-onset IBD among those over 65 poses a substantial challenge to healthcare infrastructure. It is estimated to cost nearly USD 10 billion in Canada and the US in 2018, and it is projected that this compounding burden and cost will be expected to continue, potentially exacerbating the financial strain on healthcare systems [20]. Even wealthy nations with the ability to afford costly treatment, such as biologics, would eventually reach a fiscally challenging threshold [21]. Furthermore, the indirect costs of IBD care, such as loss of productivity and income taxes, exceed the direct cost, complicating these issues [22]. In 2025, an aging IBD population will bring new challenges to the world. Therefore, the importance of introducing multiple strategies, such as increasing investments in IBD research, raising awareness, and harmonizing public along with private sector treatment plans to enhance healthcare equity, cannot be overstated.

We acknowledge that the current study also has some limitations. First, the GBD study recorded cases based on ICD-10, which is subject to misclassification that could introduce bias and possibly underestimate the true burden [23]. Second, data provided by newly industrialized countries could have missed some cases that lacked access to healthcare and complexity in diagnosing this significant but atypical presentation, which would further underestimate the case burden. Third, it is important to note that the GBD 2019 dataset does not offer detailed information on the specific impact of treatment interventions, such as surgery and biological treatments, on mortality and disability. Fourth, IBD, particularly in the elderly, is very complex and heterogeneous, which makes it challenging to visualize the rising burden. While it would be beneficial to have more granular data on the severity and classification of IBD, such detailed information is not provided within the scope of the GBD 2019 dataset. Therefore, larger epidemiological studies and predictive forecasting models are needed to fully comprehend this potentially distinctive form of IBD. Nevertheless, GBD 2019 reveals the most up-to-date global epidemiology of elderly-onset IBD, which is essential information for policymakers to learn the trend and prepare healthcare systems for the upsurge of this distinct entity.

## 5. Conclusions

In summary, elderly-onset IBD poses a public health challenge. Although rates of deaths and DALYs are decreasing globally, the high overall burden combined with the rising incidence rate could overwhelm the healthcare system and possibly lead to healthcare disparity worldwide. Effective public health intervention and sophisticated and larger epidemiological studies are warranted to understand and blunt the burden of IBD, particularly in the elderly.

## Figures and Tables

**Figure 1 jcm-12-05142-f001:**
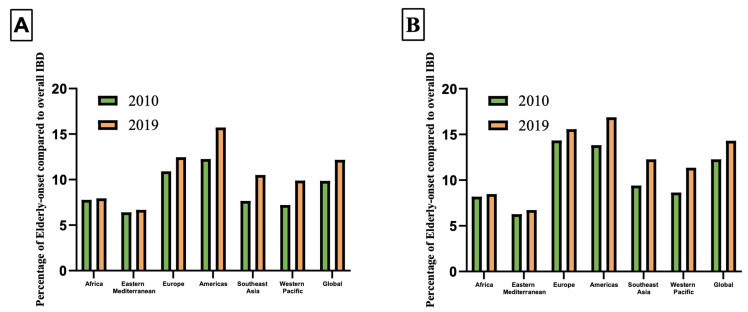
(**A**) Percentage of elderly-onset IBD compared to overall IBD among males, stratified by World Health Organization region. (**B**) Percentage of elderly-onset IBD compared to overall IBD among females, stratified by World Health Organization region. Figure legend: IBD: inflammatory bowel disease.

**Figure 2 jcm-12-05142-f002:**
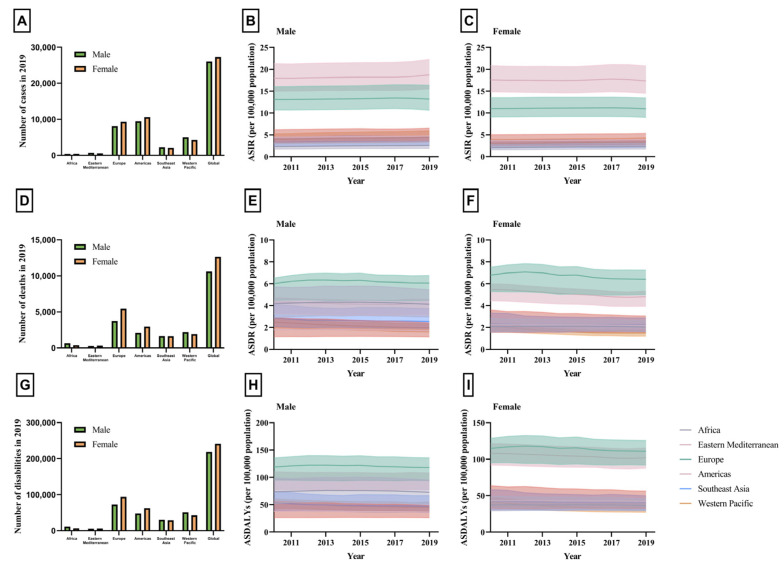
(**A**) Number of new cases of patients with IBD aged 65–89 in males versus females in 2019, stratified by World Health Organization region. (**B**) Age-standardized incidence rate of patients with IBD aged 65–89 in males from 2010 to 2019, stratified by World Health Organization region. (**C**) Age-standardized incidence rate of patients with IBD aged 65–89 in females from 2010 to 2019, stratified by World Health Organization region. (**D**) Number of deaths of patients with IBD aged 65–89 in males versus females in 2019, stratified by World Health Organization region. (**E**) Age-standardized death rate of patients with IBD aged 65–89 in males from 2010 to 2019, stratified by World Health Organization region. (**F**) Age-standardized death rate of patients with IBD aged 65–89 in females from 2010 to 2019, stratified by World Health Organization region. (**G**) Number of disabilities of patients with IBD aged 65–89 in males versus females in 2019, stratified by World Health Organization region. (**H**) Age-standardized disability-adjusted life years of patients with IBD aged 65–89 in males from 2010 to 2019, stratified by World Health Organization region. (**I**) Age-standardized disability-adjusted life years of patients with IBD aged 65–89 in females from 2010 to 2019, stratified by World Health Organization region. Figure legend: ASDALYs: age-standardized disability-adjusted life years; ASDR: age-standardized death rate; ASIR: age-standardized incidence rate.

**Figure 3 jcm-12-05142-f003:**
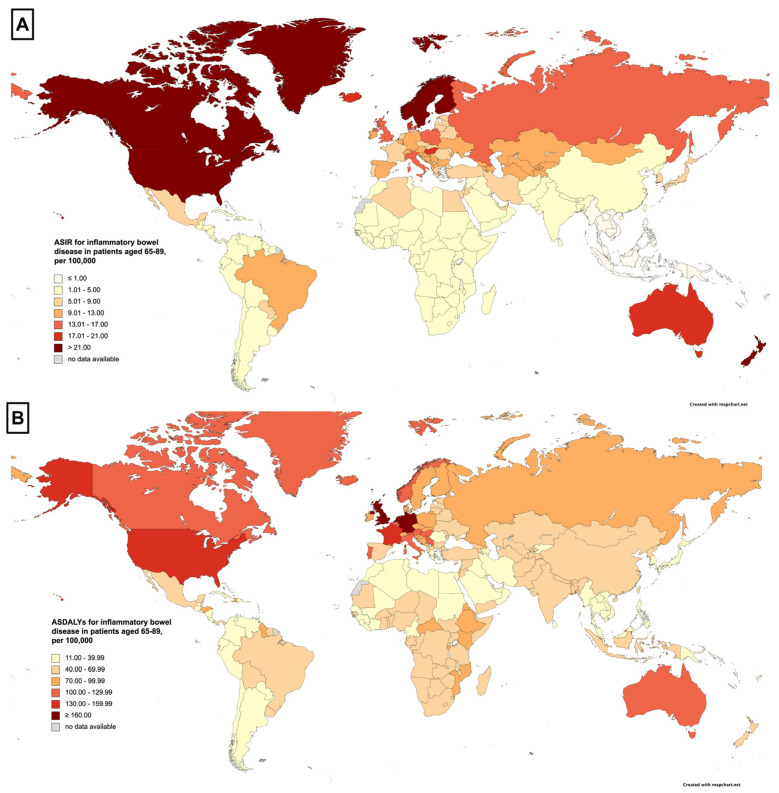
(**A**) Age-standardized incidence rate of patients with inflammatory bowel disease aged 65–89 in 2019 in each country. (**B**) Age-standardized disability-adjusted life year rate of patients with inflammatory bowel disease aged 65–89 in 2019 in each country. Figure legend: ASDALYs: age-standardized disability-adjusted life years; ASIR: age-standardized incidence rate.

**Table 1 jcm-12-05142-t001:** Incidence of inflammatory bowel disease in all ages and ages 65–89 in 2010 and 2019, and the percentage of those aged 65–89 to overall, stratified by gender.

	Males				Females			
	2010		2019		2010		2019	
	IBD in Patient Aged 65–89	Percentage of Aged 65–89 to Overall	IBD in Patient Aged 65–89	Percentage of Aged 65–89 to Overall	IBD in Patient Aged 65–89	Percentage of Aged 65–89 to Overall	IBD in Patient Aged 65–89	Percentage of Aged 65–89 to Overall
Global	18,747.87	9.85%	26,029.84	12.17%	21,109.48	12.28%	27,297.73	14.31
Region								
Africa	288.17	7.77%	412.7	7.94%	289.6	8.19%	428.55	8.47%
Eastern Mediterranean	508.19	6.4%	716.56	6.68%	382.31	6.28%	553.66	6.73%
Europe	6701.27	10.9%	8128.09	12.46%	8273.82	14.34%	9319.63	15.58%
Americas	6585.68	12.25%	9454.55	15.73%	8033.62	13.83%	10,603.66	16.88%
Southeast Asia	1502.49	7.66%	2266.15	10.51%	1390.22	9.4%	2099.30	12.27%
Western Pacific	3136.91	7.21%	5013.01	9.89%	2714.20	8.64%	4254.55	11.36%
SDI								
Low	378.22	7.26%	547.62	8.09%	348.74	7.77%	515.58	8.58%
Low-Middle	1443.08	8.38%	2035.61	10.24%	1359.06	9.6%	1971.13	11.57%
Middle	2260.48	7.85%	3637.88	10.34%	2046.41	9.01%	3294.56	11.56%
High-Middle	4709.37	10.24%	6293.83	12.43%	5931.03	14.44%	7217.30	16.41%
High	9950.22	10.7%	13,506.38	13.33%	11,417.34	12.78%	14,290.41	15.02%

Abbreviations: IBD, inflammatory bowel disease SDI, sociodemographic index.

**Table 2 jcm-12-05142-t002:** Age-standardized incidence rates of patients aged 65–89 with inflammatory bowel disease in 2010 and 2019, and the temporal changes in age-standardized incidence rates from 2010 to 2019, stratified by gender.

	Male				Female			
	2010 ASIR, per 100,000 (95% UI)	2019 ASIR, per 100,000 (95% UI)	APC in ASIR (95% UI)	*p*	2010 ASIR, per 100,000 (95% UI)	2019 ASIR, per 100,000 (95% UI)	APC in ASIR (95% CI)	*p*
Global	7.92 (6.39 to 9.71)	8.12 (6.5 to 10.04)	0.29 (0.21 to 0.36)	<0.001	7.34 (6.04 to 8.92)	7.18 (5.85 to 8.77)	−0.21 (−0.32 to −0.09)	<0.001
Region								
Africa	2.36 (1.76 to 3.21)	2.63 (1.94 to 3.52)	1.18 (1.04 to 1.32)	<0.001	2.1 (1.57 to 2.84)	2.35 (1.76 to 3.12)	1.28 (1.14 to 1.42)	<0.001
Eastern Mediterranean	4.47 (3.19 to 6.18)	4.75 (3.44 to 6.56)	0.68 (0.64 to 0.72)	<0.002	3.61 (2.61 to 5.04)	3.88 (2.83 to 5.32)	0.82 (0.79 to 0.84)	<0.001
Europe	13.11 (10.69 to 16.02)	13.2 (10.66 to 16.35)	0.14 (0.03 to 0.25)	0.014	11.02 (9.1 to 13.5)	10.98 (8.98 to 13.46)	−0.02 (−0.09 to 0.04)	0.479
Americas	17.94 (15.02 to 21.28)	18.76 (15.54 to 22.28)	0.47 (0.35 to 0.58)	<0.001	17.57 (14.84 to 20.82)	17.31 (14.5 to 20.76)	0.03 (−0.17 to 0.23)	0.75
Southeast Asia	3.28 (2.39 to 4.45)	3.52 (2.6 to 4.73)	0.75 (0.57 to 0.94)	<0.002	2.71 (2 to 3.67)	2.92 (2.18 to 3.91)	0.66 (0.36 to 0.96)	0.001
Western Pacific	3.99 (2.99 to 5.27)	4.47 (3.37 to 5.95)	1.22 (1.09 to 1.36)	<0.003	3.02 (2.32 to 3.94)	3.32 (2.53 to 4.32)	1.02 (0.93 to 1.11)	<0.001
SDI								
Low	2.79 (2.07 to 3.77)	3.03 (2.26 to 4.03)	0.92 (0.81 to 1.04)	<0.001	2.45 (1.81 to 3.24)	2.69 (2.02 to 3.57)	1.03 (0.93 to 1.14)	<0.001
Low-Middle	3.78 (2.82 to 5.02)	3.99 (2.99 to 5.29)	0.61 (0.55 to 0.66)	<0.001	3.23 (2.46 to 4.27)	3.43 (2.6 to 4.52)	0.59 (0.47 to 0.72)	<0.001
Middle	3.37 (2.51 to 4.52)	3.85 (2.85 to 5.17)	1.41 (1.31 to 1.51)	<0.001	2.72 (2.05 to 3.6)	3.07 (2.3 to 4.07)	1.32 (1.24 to 1.39)	<0.001
Middle-High	8.19 (6.59 to 10.17)	8.03 (6.37 to 10.09)	−0.2 (−0.26 to −0.14)	<0.001	7.55 (6.15 to 9.34)	7.12 (5.75 to 8.87)	−0.63 (−0.69 to −0.58)	<0.001
High	16.48 (13.67 to 19.64)	17.24 (14.24 to 20.57)	0.44 (0.37 to 0.5)	<0.001	14.79 (12.51 to 17.48)	15.13 (12.64 to 18.07)	0.3 (0.21 to 0.39)	<0.001

Abbreviations: APC, annual percentage change; ASIR, age-standardized incidence rate; CI, confidence interval; SDI, sociodemographic index, UI, uncertainty interval.

**Table 3 jcm-12-05142-t003:** Age-standardized death rates of patients aged 65–89 with inflammatory bowel disease in 2010 and 2019, and the temporal changes in age-standardized death rates from 2010 to 2019, stratified by gender.

	Male				Female			
	2010 ASDR, per 100,000 (95% UI)	2019 ASDR, per 100,000 (95% UI)	APC in ASDR (95% UI)	*p*	2010 ASDR, per 100,000 (95% UI)	2019 ASDR, per 100,000 (95% UI)	APC in ASDR (95% CI)	*p*
Global	3.65 (2.87 to 4.1)	3.31 (2.63 to 3.69)	−1.19 (−1.29 to −1.08)	<0.001	3.91 (3.19 to 4.26)	3.33 (2.79 to 3.66)	−2.02 (−2.26 to −1.79)	<0.001
Region								
Africa	4.17 (2.89 to 5.7)	4.12 (2.95 to 5.44)	−0.18 (−0.45 to 0.09)	0.191	2.08 (1.59 to 2.82)	2.03 (1.61 to 2.63)	−0.32 (−0.49 to −0.16)	<0.001
Eastern Mediterranean	1.8 (1.15 to 2.9)	1.71 (1.13 to 2.42)	−0.55 (−0.66 to −0.44)	<0.001	2.24 (1.55 to 3.59)	2.17 (1.57 to 3.04)	−0.36 (−0.48 to −0.25)	<0.001
Europe	6.01 (4.48 to 6.52)	6.06 (4.47 to 6.75)	0.06 (−0.3 to 0.42)	0.751	6.79 (5.28 to 7.51)	6.41 (5 to 7.24)	−0.82 (−1.52 to −0.11)	0.024
Americas	4.33 (3.24 to 4.7)	4.16 (3.15 to 4.57)	−0.47 (−0.73 to −0.2)	0.001	5.46 (4.44 to 5.99)	4.81 (3.93 to 5.32)	−1.56 (−1.89 to −1.23)	<0.001
Southeast Asia	2.83 (2.07 to 4.09)	2.55 (1.91 to 3.72)	−1.2 (−1.45 to −0.95)	<0.001	2.32 (1.58 to 3.31)	2.27 (1.63 to 2.86)	−0.24 (−0.52 to 0.05)	0.108
Western Pacific	2.48 (1.94 to 2.78)	1.97 (1.59 to 2.36)	−2.65 (−2.9 to −2.38)	<0.001	2.12 (1.63 to 2.41)	1.5 (1.2 to 1.79)	−3.88 (−4.14 to −3.61)	<0.001
SDI								
Low	3.77 (2.47 to 5.54)	3.63 (2.47 to 5.03)	−0.46 (−0.69 to −0.22)	<0.001	2.41 (1.72 to 3.9)	2.42 (1.81 to 3.47)	0.1 (0.03 to 0.17)	0.011
Low-Middle	3.38 (2.25 to 4.64)	3.03 (2.2 to 3.99)	−1.25 (−1.56 to −0.94)	<0.001	2.82 (1.8 to 3.89)	2.64 (1.92 to 3.23)	−0.65 (−0.88 to −0.42)	<0.001
Middle	2.41 (1.86 to 2.72)	2.07 (1.62 to 2.43)	−1.68 (−1.73 to −1.62)	<0.001	2.12 (1.64 to 2.34)	1.69 (1.38 to 1.94)	−2.41 (−2.58 to −2.23)	<0.001
Middle-High	2.94 (2.54 to 3.37)	2.48 (2.15 to 2.94)	−1.98 (−2.1 to −1.85)	<0.001	2.78 (2.39 to 3.05)	2.28 (1.96 to 2.55)	−2.41 (−2.68 to −2.14)	<0.001
High	5.85 (4.25 to 6.36)	5.73 (4.21 to 6.39)	−0.47 (−0.73 to −0.2)	0.003	7.68 (5.98 to 8.53)	6.9 (5.45 to 7.74)	−1.59 (−1.98 to −1.2)	<0.001

Abbreviations: APC, annual percentage change; ASDR, age-standardized death rate; CI, confidence interval; SDI, sociodemographic index; UI, uncertainty interval.

**Table 4 jcm-12-05142-t004:** Age-standardized disability-adjusted life years of patients aged 65–89 with inflammatory bowel disease in 2010 and 2019, and the temporal changes in age-standardized disability-adjusted life years from 2010 to 2019, stratified by gender.

	Male				Female			
	2010 ASDALYs, per 100,000 (95% UI)	2019 ASDALYs, per 100,000 (95% UI)	APC in ASDALYs (95% UI)	*p*	2010 ASDALYs, per 100,000 (95% UI)	2019 ASDALYs, per 100,000 (95% UI)	APC in ASDALYs (95% CI)	*p*
Global	74.04 (60.7 to 84.61)	68.25 (57.37 to 77.88)	−0.93 (−0.99 to −0.87)	<0.001	71.23 (60.49 to 79.72)	63.51 (54.35 to 71.74)	−1.4 (−1.54 to −1.26)	<0.001
Region								
Africa	73.75 (51.92 to 98.67)	72.79 (52.64 to 95.38)	−0.18 (−0.43 to 0.06)	0.142	37.21 (29.15 to 49.15)	36.43 (29.34 to 46.27)	−0.27 (−0.34 to −0.2)	<0.001
Eastern Mediterranean	37.99 (26.66 to 56.3)	36.54 (26.38 to 48.42)	−0.42 (−0.5 to −0.34)	<0.001	43.59 (31.66 to 63.72)	42.5 (32.49 to 56.28)	−0.26 (−0.37 to −0.16)	<0.001
Europe	118.92 (95.14 to 135.57)	117.92 (94.52 to 135.71)	−0.08 (−0.34 to 0.17)	0.524	114.82 (95.16 to 128.61)	110.91 (92.03 to 125.59)	−0.47 (−0.91 to −0.02)	0.039
Americas	96.02 (77.47 to 110.09)	94.24 (77.13 to 108.16)	−0.18 (−0.27 to −0.09)	0.001	108.01 (92 to 121.59)	101.76 (87.58 to 115.41)	−0.79 (−0.91 to −0.66)	<0.001
Southeast Asia	52.99 (39.69 to 73.66)	47.08 (35.65 to 66.41)	−1.3 (−1.5 to −1.1)	<0.001	42.7 (30.53 to 58.22)	40.43 (30.01 to 49.63)	−0.61 (−0.75 to −0.48)	<0.001
Western Pacific	52.45 (43.18 to 60.99)	45.37 (36.99 to 55.09)	−1.66 (−1.8 to −1.53)	<0.001	41.26 (33.6 to 47.42)	33.32 (27.24 to 40.1)	−2.35 (−2.55 to −2.15)	<0.001
SDI								
Low	68.6 (46.56 to 97.93)	65.44 (45.43 to 88.85)	−0.54 (−0.78 to −0.3)	<0.001	44.09 (32.28 to 67.43)	43.42 (33.31 to 60.4)	−0.15 (−0.21 to −0.09)	<0.001
Low-Middle	62.82 (43.49 to 84.62)	55.9 (41.8 to 71.59)	−1.28 (−1.45 to −1.1)	<0.001	51.65 (35 to 68.44)	48.04 (36.63 to 57.68)	−0.79 (−0.93 to −0.64)	<0.001
Middle	48.77 (39.38 to 55.83)	43.99 (35.51 to 51.53)	−1.16 (−1.19 to −1.12)	<0.001	40.41 (32.31 to 45.47)	34.43 (28.87 to 39.74)	−1.77 (−1.86 to −1.68)	<0.001
Middle-High	66.84 (57.87 to 77.54)	59.92 (50.88 to 70.95)	−1.23 (−1.31 to −1.14)	<0.001	58.73 (50.84 to 67.22)	51.95 (43.99 to 61)	−1.36 (−1.72 to −1.01)	<0.001
High	117.28 (93.13 to 134.19)	114.53 (90.7 to 131.47)	−0.31 (−0.53 to −0.1)	0.005	129.64 (107.68 to 144.51)	122.49 (102.8 to 138.25)	−0.88 (−1.13 to −0.64)	<0.001

Abbreviations: APC, annual percentage change; ASDALYs, age-standardized disability-adjusted life years; CI, confidence interval; SDI, sociodemographic index; UI, uncertainty interval.

## Data Availability

The Global Burden of Disease study in 2019 offers comprehensive data on the burden of diseases and risk factors across 204 countries and territories. Access to these data is provided by the GlobalHealth Data Exchange query tool (http://ghdx.healthdata.org/gbd-results-tool accessed on 10 February 2023), which the Institute for Health Metrics and Evaluation maintains.

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
