# Peer review of "Global Epidemiology and Burden of Elderly-Onset Inflammatory Bowel Disease: A Decade in Review"

_jcm, 2023, doi:10.3390/jcm12155142_

Round 1
Reviewer 1 Report
In this review, the authors analyzed trends in elderly-onset IBD over a decade encompassing the estimation of annual frequencies and age-standardized rates (ASRs) of elderly-onset IBD burden for both genders, stratifying by geographical and sociodemographic factors.
The authors emphasized that a notable increase in the proportion of elderly-onset IBD, around 13% of all IBD cases. They observed a rising incidence in males, contrasted by a decreasing trend in females. In this study, the highest surge in incidence rates was seen in the Western Pacific region in both genders, but the highest burden was observed in America.
This study is well written and will provide valuable information to our knowledge. However, I do have minor suggestion.
Abbreviations should be defined the first time. In abstract, there was no first definition of both SDI and DALYs. Additionally, the acronym "ASRs" existed but was not used again.
In the discussion section, the authors stated some limitations of the current study. There are also another limitation that should be included in the study is the necessity of biological therapy and abdominal surgery (colectomy or intestinal resection).
Author Response
Global Epidemiology and Burden of Elderly-Onset Inflammatory Bowel Disease: A Decade in Review
We express our sincerest gratitude to the editors and reviewers for their valuable and constructive feedback on our manuscript. The insightful comments and suggestions have greatly enhanced our understanding of the complexities surrounding the elderly-onset inflammatory bowel disease which is rapidly increasing globally. We have given careful consideration to all the feedback and have made significant revisions to the manuscript to incorporate these suggestions. We are confident that these changes have greatly strengthened the overall quality of our paper. We sincerely appreciate the time and effort invested by the reviewers and editors in carefully assessing our work and their valuable contributions in improving our manuscript. The specific responses to individual comments are noted below:
REVIEWER 1 COMMENTS
Comment 1: Abbreviations should be defined the first time. In abstract, there was no first definition of both SDI and DALYs. Additionally, the acronym "ASRs" existed but was not used again.
Respond: We sincerely appreciate the reviewer's valuable feedback. We acknowledge the importance of providing clear definitions to ensure the readability and understanding of our manuscript. In response to your comment, we have now included the definitions of SDI (Socio-Demographic Index) and DALYs (Disability-Adjusted Life Years) in the abstract to ensure clarity for readers. Furthermore, we have carefully reviewed the manuscript to ensure that all abbreviations, including "ASRs," are consistently defined upon their first use and subsequently used consistently throughout the text.
Comment 2: In the discussion section, the authors stated some limitations of the current study. There are also another limitation that should be included in the study is the necessity of biological therapy and abdominal surgery (colectomy or intestinal resection).
Respond: (Line 319-321, Page 14) We appreciate the reviewer's careful examination of our manuscript and the valuable suggestion regarding the limitations of our study. In the discussion section, we have highlighted some of the limitations of our study, including the lack of detailed information on burden after specific treatment interventions. We acknowledge that the impact of biological therapy and abdominal surgery on mortality and disability is an important factor in understanding the burden of elderly-onset IBD. Your feedback has been instrumental in strengthening the overall quality of our research, and we sincerely thank you for your thoughtful input.

Reviewer 2 Report
This is a nice, comprehensive look at elderly IBD worldwide using the Global Burden of Disease Study database. Important caveats are acknowledged with this methodology. A lot of data is presented and Table 1 could be better formatted for ease of interpretation.
Line 48: should spell out SDI as first use of this abbreviation
The findings of this manuscript are important.
Author Response
Global Epidemiology and Burden of Elderly-Onset Inflammatory Bowel Disease: A Decade in Review
We express our sincerest gratitude to the editors and reviewers for their valuable and constructive feedback on our manuscript. The insightful comments and suggestions have greatly enhanced our understanding of the complexities surrounding the elderly-onset inflammatory bowel disease which is rapidly increasing globally. We have given careful consideration to all the feedback and have made significant revisions to the manuscript to incorporate these suggestions. We are confident that these changes have greatly strengthened the overall quality of our paper. We sincerely appreciate the time and effort invested by the reviewers and editors in carefully assessing our work and their valuable contributions in improving our manuscript. The specific responses to individual comments are noted below:
REVIEWER 2 COMMENTS
Comment 1: A lot of data is presented and Table 1 could be better formatted for ease of interpretation.
Response: (Line 169, Page 5) We thank the reviewer for their comments and appreciate their efforts in providing constructive feedback. In response to your valuable input and in our commitment to improving the clarity and ease of interpretation of the data, we proactively made changes to Table 1. By including percentage values and removing the number of cases of IBD in the overall table, we aimed to enhance the table's readability and facilitate a better understanding of the key findings. We value the reviewer's thorough evaluation of our manuscript and are glad to have addressed your concerns.
Comment 2: Line 48: should spell out SDI as first use of this abbreviation okay please respond to the reviewer comment
Response: We would like to express our appreciation to the reviewer for their valuable observation. Indeed, you are right, and we apologize for the oversight. In response to your feedback, we have addressed the issue and spelled out "SDI" as "Socio-Demographic Index" in its first use at Line 48 of the manuscript.
